# Transcriptomic analysis reveals effects of fertilization towards growth and quality of *Fritillariae thunbergii* bulbus

Luman Huang[1], Shuang Liang[1], Lei Luo[2], Mengmin Wu[1], Hongwei Fu[1]*, Zhuoheng Zhong[1]*

**1** College of Life Sciences and Medicine, Zhejiang Sci-Tech University, Hangzhou, P.R. China, **2** Zhejiang Institute for Food and Drug Control, Hangzhou, P.R. China

* fhw668@zju.edu.cn (HF); zhongzhh@zstu.edu.cn (ZZ)

**Data Availability Statement:** The raw data of RNA-Seq have been submitted to the the NCBI Sequence Read Archive (SRA) database (Accession: PRJNA950562).

## Abstract

*Fritillariae thunbergii* Bulbus (FTB) is a traditional Chinese medicine that has been widely cultivated for its expectorant, antitussive, antiasthmatic, antiviral, and anticancer properties. The yield and quality of *F. thunbergii* are influenced by cultivation conditions, such as the use of fertilizers. However, the optimal type of fertilizers for maximum quality and yield and underlying mechanisms are not clear. We collected *F. thunbergii* using raw chicken manure (RC), organic fertilizer (OF), and plant ash (PA) as the base fertilizer in Pan'an County, Jinhua City, Zhejiang Province as experimental materials. The combined results of HPLC-ELSD detection and yield statistics showed that the *F. thunbergii* with OF application was the best, with the content of peimine and peiminine reaching 0.0603% and 0.0502%, respectively. In addition, the yield was 2.70 kg/m$^2$. Transcriptome analysis indicated that up-regulation of the ABA signaling pathway might promote bulb yield. Furthermore, putative key genes responsible for steroidal alkaloid accumulation were identified. These results provided guiding significance for the rational fertilization conditions of *F. thunbergii* as well as the basis for the exploration of functional genes related to the alkaloid biosynthesis pathway.

## Introduction

*Fritillaria* ("Beimu" in Chinese) genus, belonging to the *Liliaceae* family, was ranked as one kind of most traditional medicinal plants in China [1]. So far, about 130 species of this genus are found to be distributed in temperate regions of the northern hemisphere [2]. In the 2020 edition of the Chinese Pharmacopoeia, the bulbus of 11 cultivars from *Fritillaria* were officially recorded and classified into five species, namely *Fritillariae Cirrhosae* Bulbus (FCB), *Fritillariae Hupehensis* Bulbus (FHB), *Fritillariae Ussuriensis* Bulbus (FUB), *Fritillariae Pallidiflorae* Bulbus (FPB), and *Fritillariae Thunbergii* Bulbus (FTB) [3]. *F. thunbergii*, also known as Zhe Bei, is one of the famous "Zhejiang Eight Flavors". Modern medical research reported that it had functions such as cough relieving, phlegm relieving, asthma relieving, analgesic and anti-inflammatory, anti-ulcer, antioxidant, anti-tumor, and reversing tumor cell resistance [4–

**Funding:** Present work was supported by the Research Initiation Funding of Zhejiang Sci-Tech University (Grant number: 19042112-Y; 22042024-Y).

**Competing interests:** The authors declare that they have no known competing financial interests or personal relationships that could have appeared to influence this work reported in this paper.

8]. Its main chemical components include alkaloids, polysaccharides, total saponins, flavonoids, and volatile components [7]. According to research, steroidal alkaloids have important medicinal value in eliminating phlegm and relieving cough, reducing blood pressure and promoting blood circulation, relieving pain and ulcers, anti-inflammatory and antioxidant effects, and anti-tumor effects [9]. At present, the synthesis pathway of steroidal alkaloids in *F. thunbergii* is not very clear.

With the widespread application of FTB, wild resources are scarce. To meet the increasing market demand, it has been introduced as a domestic variety, and the planting areas are constantly expanding. Pan'an County is one of the main production areas of *F. thunbergii* [10]. However, differences in fertilization conditions, soil, and other factors have led to significant differences in the quality and yield of *F. thunbergii*. The "Technical Regulations for the Production of *F. thunbergii*" stipulated that raw chicken manure, commercial organic fertilizer, and plant ash could be used as fertilizer for *F. thunbergii*. However, there is still a lack of research on these three fertilizer applications on the yield and quality of *F. thunbergii*.

Transcriptomic information from many medicinal herbs has been acquired for investigation of differential alterations at the gene expression level as well as potential biosynthetic pathways of secondary metabolites [11]. According to previous analysis of transcriptome data, the steroid alkaloid biosynthetic pathway was divided into three stages: (1) Terpenoid backbone biosynthesis, where isopentenyl diphosphate (IPP) and dimethylallyl diphosphate (DMAPP) produced from mevalonate (MVA) and 2-C-methyl-D-erythritol-4-phosphate (MEP) pathways, respectively, were converted into farnesyl pyrophosphate (FPP) under the action of farnesyl pyrophaophate synthase (FPS) [12]; (2) Steroid circular skeleton precursor biosynthesis, two FPPs were catalyzed by squalene synthase (SQS) and squalene epoxidase (SQE) to form 2,3-oxidosqualene (OS) [13]; (3) Steroid compounds biosynthesis, several members of plant cytochrome P450 (CYP450) had also been confirmed to participate in the biosynthesis of steroid compounds [14], but the specific functions have yet to be testified. The varied enzymes involved in the biosynthesis of steroid alkaloids in FTB still need to be further studied.

In addition, many researchs have also proved that altitude have an significant impact on secondary metabolism in medicinal plants [15, 16]. Indeed, omics have been applied to the in vitro reproduction techniques of high-value Himalayan *Fritillaria* species [17]. Similarly, appropriate organic fertilizer was beneficial for the accumulation of secondary metabolites of *Salvia miltiorrhiza Bunge* and the improvement of yield [18], different fertilization conditions have an impact on the yield and quality of *Ligusticum chuanxiong Hort* [19], *Herba Artimisiae Sieversianae* [20]. However, there is little research on the effects of different fertilization conditions on the yield and alkaloid content of Bulbs of *F. thunbergii*. In this study, we collected FTB obtained with three kinds of fertilizer from Pan'an County in Zhejiang province in China. High-performance liquid chromatography-evaporative light scattering detector (HPLC-ELSD) and high-throughput RNA-Seq technology were performed to jointly analyze variations among different groups at both metabolite and gene expression levels, with the hypothesis of potential genes related to steroidal alkaloid biosynthesis pathway in *F. thunbergii* and providing a genetic basis for subsequent research on steroid alkaloid biosynthesis. It has a guiding significance for rational fertilization of *F. thunbergii* to improve yield and quality.

## Materials and methods

### Plant materials

Triennial *Fritillariae thunbergii* bulbs (cultivar 'Zhebei 1', identified by Prof. Bizeng Mao from Zhejiang University) were cultivated and collected on the same day in a natural habitat with fertile soil in Pan'an, Zhejiang, China (located at 28˚99'N, 120˚62'E, altitude 600 m). Raw

chicken manure (RC), organic fertilizer (OF), and plant ash (PA) were separately applied as the base fertilizer and the applications were maintained constant during cultivation. *Fritillaria thunbergii* bulb yield was determined by harvesting the plants in an area of 2 m$^2$ in each plot. For collecting samples for analysis, all the bulbs were collected biological replicates to randomly select different regions in the north, south, east, and west of the same production area, and mix well on the same day and washed with pure water, after which one portion was quickly frozen in liquid nitrogen and stored at -80˚C until processing, and these samples were used for RNA extraction and library construction. For each fertilization conditions, three biological replicas were prepared for each sample.

## Soil and fertilizer sources

The soil is sandy loam soil, with a turtle-shaped border. Organic fertilizer (SITEWO, Anhui, China). Chicken manure and plant ash were made by local farmers. The fertilization amount for different fertilizers was 5 kg/m$^2$.

## Chemical reagents

Strong ammonia solution (28%, Nanjing Chemical Reagent Co., Ltd), Chloroform (≥99.5%, Nanjing Chemical Reagent Co., Ltd), Methanol (≥99.5%, Hangzhou Gaojing Fine Chemical Co., Ltd).

## Determination of alkaloid contents

A method based on the Chinese Pharmacopoeia was used to extract peimine and peiminine from FTB. Briefly, dried bulbs were ground into powder and screened through a 65-mesh sieve, 2 g of the dried powder was accurately weighed into a flask (with three biological replicates per sample), soaked in 4 mL strong ammonia solution for 1 h, and accurately added with 40 mL of a mixed solution of chloroform-methanol (4:1, v/v). The mixture was heated and refluxed in a water bath at 80˚C for 2 h. Precisely measured supernatant (10 mL) was put in an evaporating dish and evaporated to dryness, 2 mL of methanol was added to dissolve the residue and accurately transferred to a 2 mL-volumetric-flask. Then the solution was centrifuged for 10 min at 13 400 *g*, and the supernatant was collected and filtrated through 0.22 μm membrane filters (Jinteng, Tianjin, China) before HPLC-ELSD analysis.

An aliquot of the distinct filtrates (10 μL) was injected into an LC-20AT HPLC system equipped with an Alltech 2000ES ELSD Evaporative Light Scattering Detector and an automatic sampler (Shimadzu, Japan). The chromatographic separation was applied on a kromasil C18 column (250 mm × 4.6 mm, 5 μm) at the column temperature of 35˚C and the flow rate is 1.0 mL/min. The mobile phase consisted of methanol (A) and 0.03% diethylamine solution (B) for chromatographic separation using isocratic elution parameters.

## Total RNA preparation

For RNA-seq of FTB from different fertilization conditions, the total RNA was extracted using the *SteadyPure* Plant RNA Extraction Kit (Accurate Biotechnology, Hunan, China). The integrity and purity of the total RNA quality were determined by the PowerPac™ Basic electrophoresis apparatus (Bio-Rad, California, USA), and the content was quantified using the NanoDrop spectrophotometer 1000 (Thermo Fisher, MA, USA). Samples from three fertilization conditions containing three replicates were applied for the construction sequencing library.

## Library construction and sequencing

Library construction and sequencing were performed by Novogene Bioinformatics Technology Co. Ltd. (Beijing, China). The extracted total RNA as input material and reverse-transcribed to cDNA for building library by NEBNext® Ultra™ RNA Library Prep Kit for Illumina® (New England Biolabs, Beijing, China) following the manufacturer's protocol. The modified cDNA fragments with a length of 370~420 bp were preferentially selected, and amplified by PCR, then the PCR products were purified using the AMPure XP beads (Beckman Coulter, Beverly, USA) to obtain cDNA libraries from different conditions of *F. Thunbergii* plants. Finally, the Illumina novaseq 6000 platform was used for transcriptome sequencing, and 150 bp paired-end reads were generated by paired-end sequencing technology.

## Transcriptome assembly and annotation

For assessment of the quality and reliability of the transcriptome sequencing data, the raw data was filtered to remove polynucleotide sequences, reads with adapters, and reads with nebulous bases. Then the high-quality clean reads were *de novo* assembled using Trinity (v2.6.6). Based on Trinity splicing, Corset software was used to aggregate redundant transcripts for a better detection rate of differentially expressed genes [21]. BUSCO software was used to evaluate the splicing quality of the spliced files [22]. Finally, all assembled transcripts were mapped to public databases by RSEM (v12.15) [23], and the database contains NCBI non-redundant protein sequences (Nr), NCBI non-redundant nucleotide sequences (Nt), Protein family (Pfam), euKaryotic Orthologous Groups (KOG), Swiss-Prot, Kyoto Encyclopedia of Genes and Genomes (KEGG) and Gene Ontology (GO) databases.

## Differentially expressed genes analysis

The relative expression levels of each transcript were calculated according to fragments per kilobase of transcript per million mapped reads (FPKM) [24]. Before screening the differentially expressed genes (DEGs), unigenes with unstable expression in sample transcriptomes were removed. The FPKM value of three samples with $p$-value < 0.05 was set as the threshold for differentially expressed genes (DEGs). Furthermore, GOseq (1.10.0) and KOBAS (v2.0.12) software were used for GO function enrichment analysis and KEGG metabolic pathway enrichment analysis of DEGs, respectively [25, 26]. Furthermore, the clustered expression pattern was drawn through R (Version 3.0.3), Package ggplot2 and Package pheatmap.

## Quantitative real time-polymerase chain reaction (qRT-PCR) analysis

To confirm the accuracy of the gene expression levels of DEGs obtained by transcript sequencing, qRT-PCR analysis was used. The primers used for qRT-PCR are shown in S1 Table. Total RNA was reverse-transcribed into cDNA using the *Evo M-MLV* RT Mix kit (Accurate Biotechnology, Hunan, China), then cDNA was used as the template for qRT-PCR using the SYBR® Green Premix *Pro Taq* HS q-PCR kit (Accurate Biotechnology, Hunan, China) on Applied BiosystemsTM 7500 Fast Dx Real-Time PCR Instrument (ABI, Carlsbad, USA). *Actin* was used as the reference gene, and the relative expression levels of each group (containing three biological replicates) were calculated based on the $2^{-\Delta\Delta Ct}$ method [27].

## Statistical analysis

All statistical analyses including steroid alkaloid contents and relative gene expressions were performed One-way ANOVA using SPSS (25.0.0).

**Table 1. Contents of two steroidal alkaloids in FTB from different fertilization conditions.**

| Sample | Peimine | Peiminine | Total content% | Production(Kg/m²) |
|---|---|---|---|---|
| | Content percentage% | Content percentage% | | |
| RC | 0.0452 ± 0.001[c] | 0.0362 ± 0.0012[c] | 0.0814 ± 0.0022[c] | 2.72 |
| OF | 0.0603 ± 0.0008[b] | 0.0502 ± 0.0012[b] | 0.1105 ± 0.002[b] | 2.70 |
| PA | 0.0713 ± 0.0011[a] | 0.0615 ± 0.0018[a] | 0.1328 ± 0.0031[a] | 2.13 |

Note.

[a], [b] and [c] indicate significant differences in alkaloid content between different fertilization conditions ($P < 0.05$).

## Results

### The contents of peimine and peiminine varied in FTB from different fertilization conditions

As the steroidal alkaloid peimine and peiminine are important medicinal ingredients in FTB, the amount of the two alkaloids in different fertilization conditions was quantified based on HPLC-ELSD analysis. As shown in Table 1, the contents of the two alkaloids in PA were the highest, with the content of peimine and peiminine were 0.0713% and 0.0615%, respectively. However, compared with the other two groups, the yield was the lowest (2.13 Kg/m²). The content of the two alkaloids in RC was the lowest, which just reached the pharmacopoeia standard (The total quantity of peimine and peiminine based on dried material content serves as a crucial quality control standard, with a minimum requirement of 0.080%.), but the yield was the highest (2.72 Kg/m²). The results indicated that, apart from RC which had a low concentration of pharmacologically active compounds, other FTBs produced in two conditions of Zhejiang province were excellent. Interestingly, obtained through comprehensive evaluations, the optimal fertilization condition of *F. thunbergii* was OF.

### Transcriptome *de novo* assembly and functional annotations of unigenes

To investigate the difference of FTB from three fertilization conditions at the transcriptomic level, 9 RNA-seq libraries of FTB from three groups each with three biological replicates were constructed. Each library obtained an average of approximately 21 to 22 million raw reads. After the removal of adapters, uncertain bases, and low-quality raw sequences, a total of 328.14 million clean reads of FTB were obtained, ranging from 20.13 million to 23.27 million for each library. The Q20 and Q30 values (the percentage of bases with quality values ≥ 20 or 30, which refers to the error probability of the identified bases in the base calling process and used to measure the sequencing accuracy) of the nucleotide sequences reached above 90%, and the GC contents of each sample clean reads were around 50% (S2 Table). PCA analysis showed that the transcriptome data within the sample group had good repeatability (S1 Fig).

After *de novo* assembly of clean reads from 9 samples of FTB using Trinity software, a total of 139, 639 Unigenes were obtained through Corset clustering for subsequent comparative analysis. Unigenes with sequence lengths ranging from 300 bp to 500 bp, from 500 bp to 1000 bp, from 1000 bp to 2000 bp, and greater than 2000 bp exhibit a gradual decrease in the number of unigenes within the corresponding interval as the sequence length interval increases (S3 Table).

All the assembled unigenes were annotated according to NR, NT, Swiss-Prot, Pfam, GO, KOG and KEGG databases. The total number of unigenes with at least one annotation was 59, 428, making up approximately 50% of all unigenes. Among them, 7,400 unigenes were

**Table 2. Unigenes functional annotation of the FTB transcriptome.**

| Database | Unigenes | Percentage% |
|---|---|---|
| NR | 47853 | 41.53% |
| NT | 24581 | 21.33% |
| Swiss-Prot | 35293 | 30.63% |
| PFAM | 33951 | 29.47% |
| GO | 33948 | 29.46% |
| KOG | 11079 | 9.62% |
| KEGG | 17286 | 15.00% |
| All Database-Annotated | 7400 | 6.42% |
| At least one Database-Annotated | 59428 | 51.58% |

functionally annotated in the seven databases (Table 2). 47, 853 unigenes were annotated at most in Nr database, and Nr homologous species distribution analysis indicated that the highly homologous species related to *F. thunbergii*, such as *Elaeis guineensis* (6, 272; 13.11%), *Phoenix dactylifera* (5, 926; 12.38%), *Quercus suber* (5, 743; 11.09%), and *Asparagus officinalis* (3, 303; 6.90%) (S2A Fig). According to GO clustering analysis, 33, 948 unigenes were annotated (S2B Fig).

To identify involved biological pathways of the transcripts, 17, 286 unigenes were identified in 351 KEGG pathways with the most abundant number in metabolic pathways (S2C Fig). Among the metabolic pathways, several unigenes of secondary metabolic biosynthetic pathways might be involved in regulating the biosynthesis of steroid alkaloids, including 134 unigenes of terpenoid backbone biosynthesis (ko00900), 16 unigenes of sesquiterpenoid and triterpenoid biosynthesis (ko00909), 68 unigenes of steroid biosynthesis (ko00100), 47 unigenes of diterpenoid biosynthesis (ko00904) and 81 unigenes of ubiquinone and other terpenoid-quinone biosynthesis (ko00130).

## Analysis of differentially expressed genes (DEGs)

A total of 16, 810 DEGS were screened according to the FPKM of three groups with $p$-value < 0.05. Firstly, we analyzed these DEGs based on GO annotation. The statistical analysis results are shown (Fig 1), where differentially expressed genes are distributed in three major categories: "biological process", "cellular component", and" molecular function". According to the P-value enriched by GO, it was found that in the category of "biological processes", the enrichment degree of "protein metabolic process" (1637) and "ribonucleoprotein complex biogenesis" (375) was the most significant. In the category of "Cellular component", the enrichment degree of "ribonucleoprotein complex" (434) and "macromolecular complex" (1923) is the most significant and the main taxonomic unit. In the category of "Molecular function", the enrichment level of "protein binding function" (1157) and "RNA binding function" (316) is the most significant. The results indicate that the GO subclass with the most significant enrichment of differential genes is mainly related to protein function and the binding function of various genetic materials.

KEGG annotation analysis illustrated that DEGs were mainly related to pathways such as terpenoid backbone biosynthesis (ko00900), sulfur metabolism (ko00920), isoquinoline alkaloid biosynthesis (ko00950), steroid biosynthesis (ko00100), stilbenoid, diarylheptanoid and gingerol biosynthesis (ko00945), tropane, piperidine and pyridine alkaloid biosynthesis (ko00960), carotenoid biosynthesis (ko00906), and zeatin biosynthesis (ko00908), which

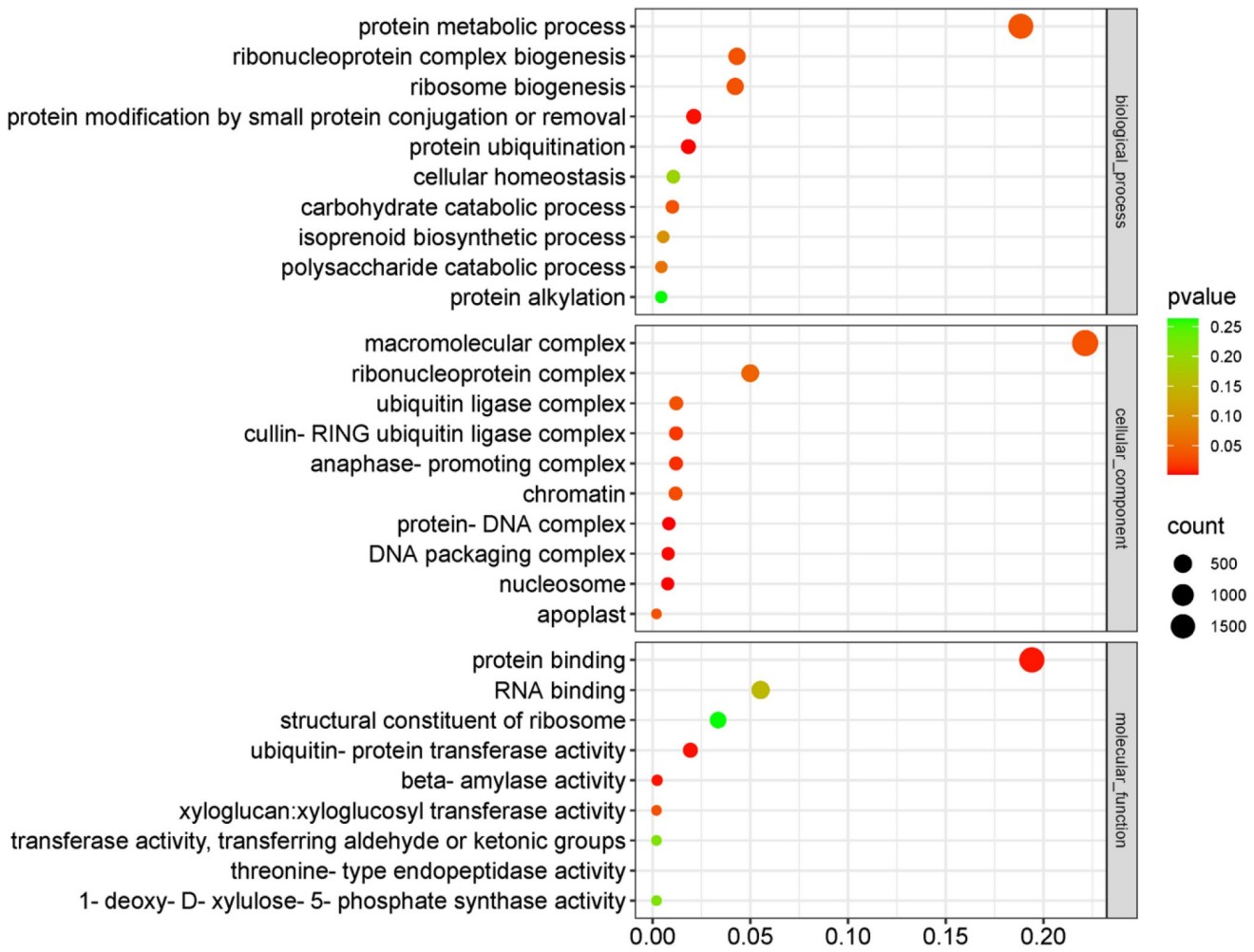

**Fig 1. GO functional classification of total DEGs.** The y-axis indicates the different categories. The x-axis indicates the gene ratio of enrichment analysis, The size of circle indicates the number of differential genes. The color of circle indicates the P-value.

indicated that expression levels of genes related to secondary metabolic pathways in different FTBs were discrepant (Fig 2).

Next, five enrichment pathways with the highest number of DEGs were selected and involved in regulating plant life activities. The plant hormone signal transduction pathway (ko04075) in the "Environmental Information Processing" branch is an important pathway for regulating plant growth [28] (Fig 3A, S4 Table). The "plant pathogen interaction pathway (ko04626)" in the "organic system" branch mainly includes immune responses activated by pathogen-related molecular features and immune responses activated by effector proteins [29] (Fig 3B, S5 Table). The plant immune system can resist the invasion of most natural pathogens through these two defense mechanisms [30]. In the "genetic information processing" branch, the "mRNA monitoring pathway (ko03015)" is also known as uninterrupted decay (Fig 3C, S6 Table), which can effectively prevent the synthesis of other abnormal proteins while eliminating some harmful effects [31]; the "Ubiquitin mediated protein hydrolysis (ko04120)" is the main pathway for intracellular protein degradation (Fig 3D, S7 Table) and participates in multiple processes throughout the cell life cycle to regulate protein stability, function, activity, and

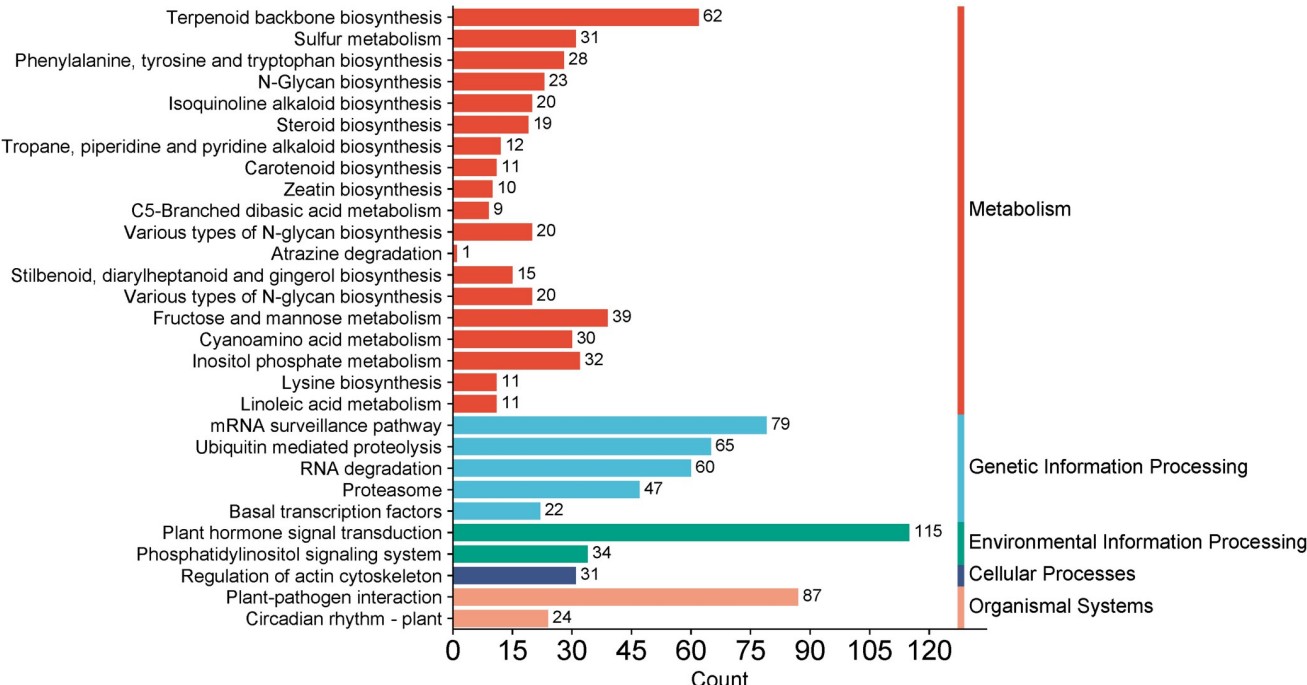

**Fig 2. KEGG functional classification of total DEGs.** The y-axis indicates the different categories. The x-axis indicates the number of DEGs in a category.

subcellular localization [32, 33]; The "terpenoid skeleton biosynthesis (ko00900)" is involved in the generation of steroid alkaloid precursors (Fig 3E, S8 Table) and is highly important for steroid alkaloid synthesis [34].

## Pathway mapping analysis of DEGs involved in plant hormone signal transduction

Plant hormones are small plant organic molecules, which mainly contain auxin, cytokinin (CK), gibberellin (GA), abscisic acid (ABA), ethylene (ET), brassinosteroid (BR), jasmonic acid (JA), salicylic acid (SA) and strigolactones. They play an indispensable role in regulating plant growth and abiotic stress responses at very low concentrations in plants [35]. Based on KEGG annotation and gene expression files, a total of 123 DEGs were mapped to the plant hormone signal transduction pathway (ko04075) (Fig 4, S4 Table).

ABA is the prominent medium of these response processes, the gene expression of *SnRK2* and *JAZ* in PA was the highest. In addition, *ABF*, as a master regulatory transcription factor, belongs to a subfamily of bZIP proteins [36] and is also up-regulated in PA. In the JA signaling pathway, the expression profile of *JAZ* in PA and OF was relatively higher, Similarly, the majority of transcripts enriched in the JA signaling pathway were increased in PA, which was Similar to the expression patterns in the ABA pathway (Fig 4).

## Identification and expression analysis of DEGs associated with steroid biosynthesis

To distinguish the presumable genes involved in the steroidal alkaloids biosynthesis in *F. thunbergii*, a possible biosynthetic pathway containing DEGs expression level has been made according to all DEGs enrichment results of KEGG pathways, it was initially synthesized via

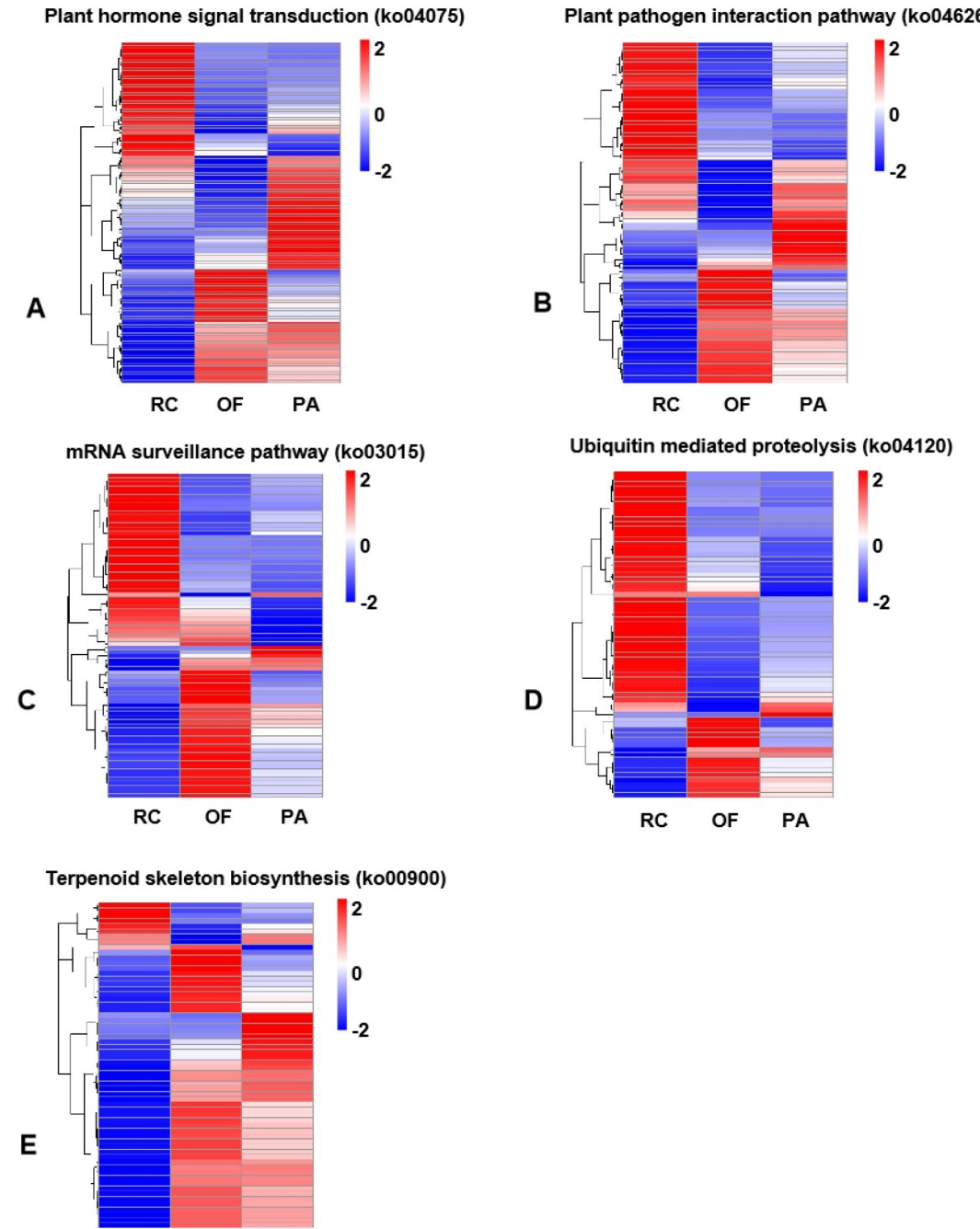

**Fig 3. The clustered expression profiles of five enriched pathways with the highest number of DEGs in the FTB of three fertilization conditions.** FPKM of each gene were normalized according to Z-score. Red indicates high gene expression, blue indicates low gene expression.

terpenoid backbone biosynthesis (ko00900) followed by steroid biosynthesis (ko00100). The results indicated that 62 DEGs were involved in terpenoid backbone biosynthesis (ko00900). Interestingly, these DEGs mainly show the same expression trend: low expression level in RC, and the expression level increases successively in PA and OF (Fig 5, S8 Table). which is

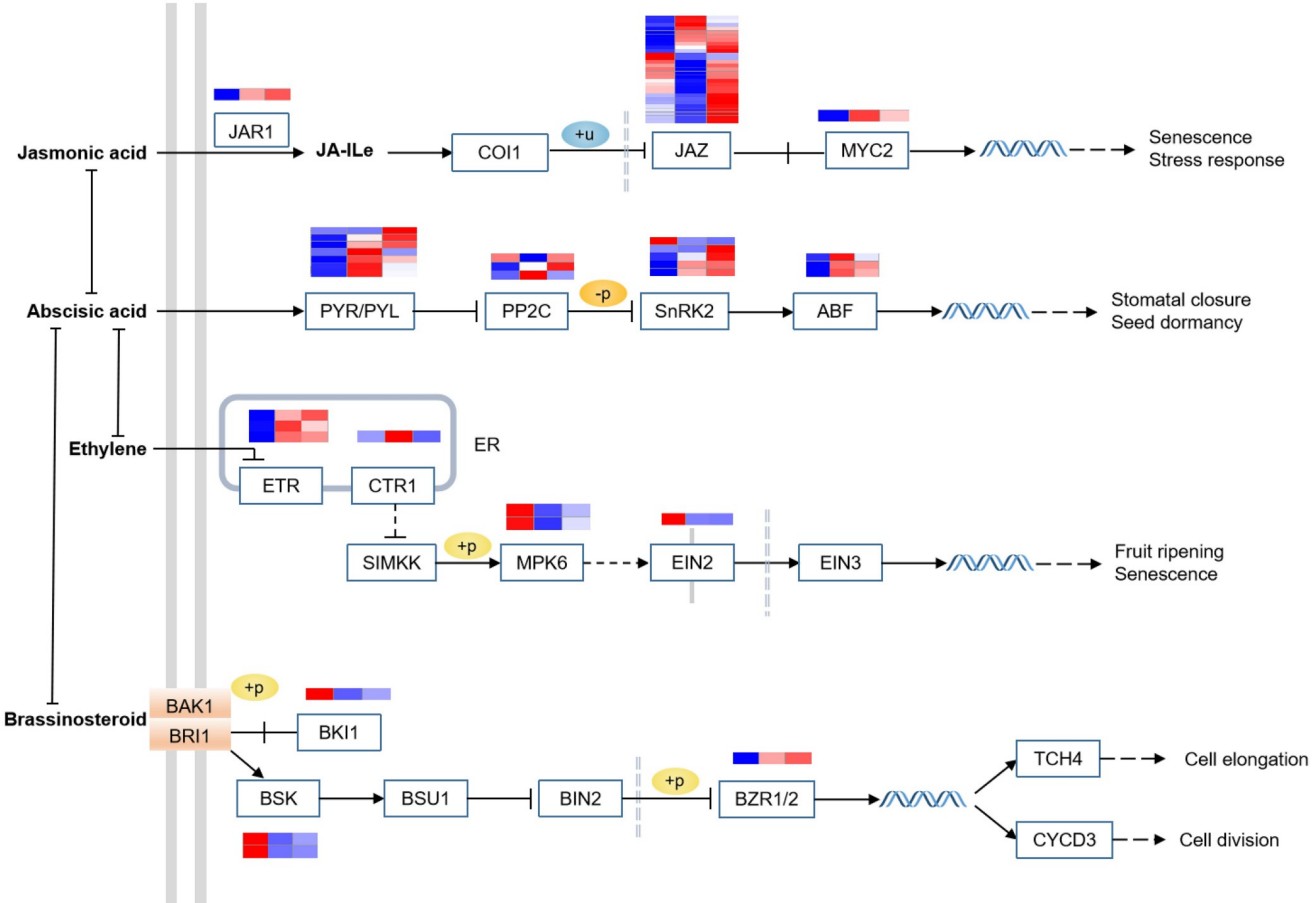

**Fig 4. Expression heat maps of DEGs for the structural genes involved in plant hormone signal transduction (ko04075) from three fertilization conditions.** FPKM of each gene were normalized according to Z-score. Red indicates high gene expression, blue indicates low gene expression. Detailed gene expression profiles were shown in S4 Table.

consistent with the trend of differences in FTB content. DXS can catalyze the production of 1-deoxyglucose 5-phosphate (DXP) from pyruvate and 3-phosphate glyceraldehyde, which is the first rate limiting step in the catalytic MEP pathway. DXR can catalyze the synthesis of various important intermediate metabolite enzymes such as IPP, pyridoxine, and thiamine by DXP. Which is the second-rate limiting enzyme in the MEP pathway. Compared with RC, *DXS* and *DXR* in PA and OF were significantly upregulated.

According to a previous study, cholesterol was the precursor for the biosynthesis of steroidal alkaloids, which is important for the downstream accumulation of steroid alkaloids [34, 37]. As shown in Fig 6 (S9 Table), the pathway from 2,3-Oxidosqualene toward cholesterol in the KEGG database was already clear, we can directly obtain relevant information on DEGs in this pathway through enrichment analysis. the expression level of DEGs mainly is up-regulated in OF and PA, and a downregulation trend in RC. To some extent, it provides reliable evidence for the upregulation of genes related to steroid biosynthesis pathways.

Peimine and peiminine are considered to be the principal steroidal alkaloids of FTB. Yet, there is limited information about the genes involved in the downstream biosynthesis from cholesterol to steroidal alkaloids in plants at the biochemical level. The biosynthetic pathway

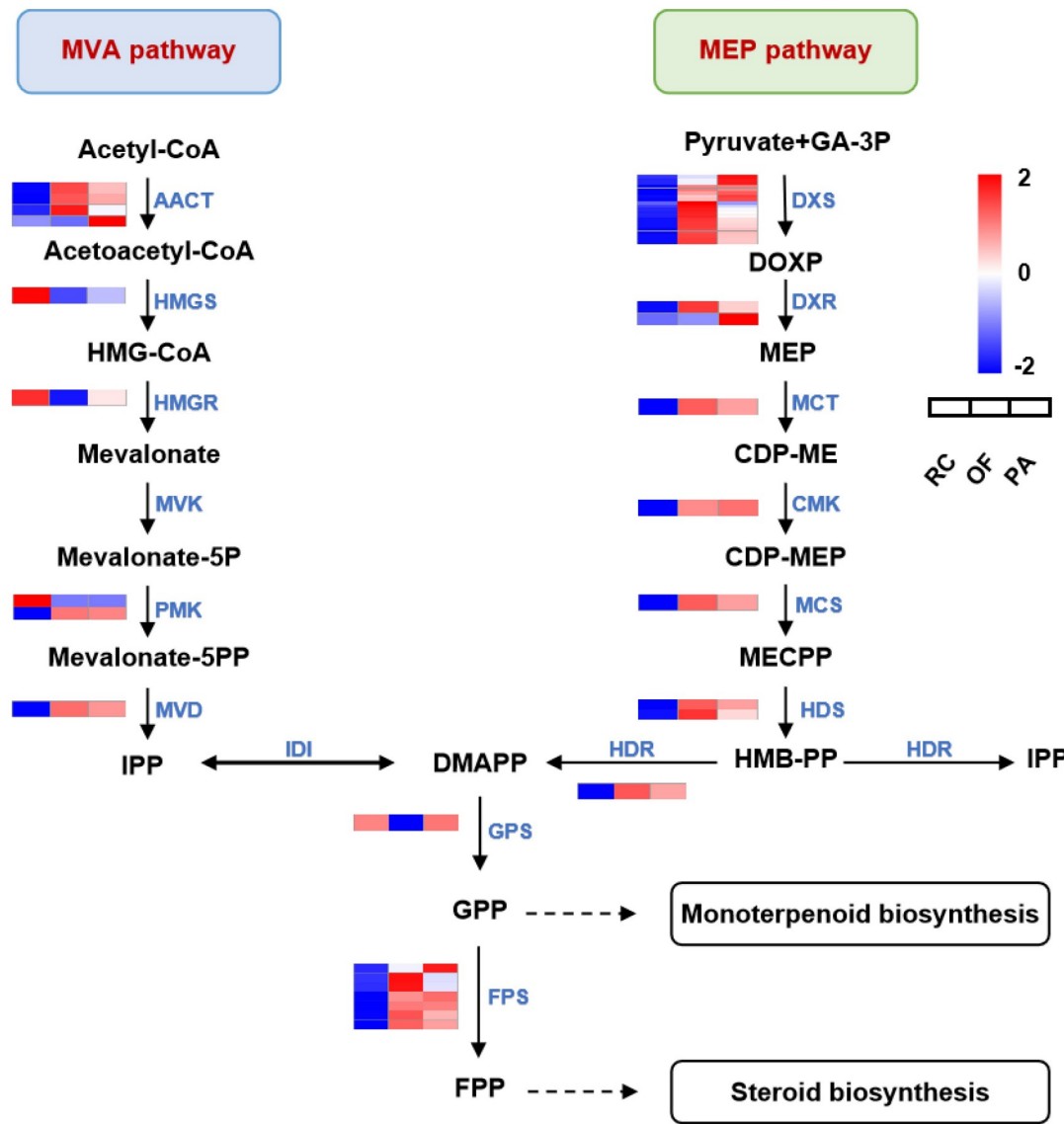

**Fig 5. Expression heat maps of DEGs in terpenoid backbone biosynthesis (ko00900) in the FTB from three fertilization conditions.** FPKM of each gene were normalized according to Z-score. The dashed arrow represents presumed terminal biosynthesis steps. AACT, acetyl CoA acetyltransferase; HMGS, 3-hydroxy-3-methylglutaryl-CoA synthase; HMGR,3-hydroxy-3-methylglutaryl-CoA reductase; MVK, mevalonate kinase; PMK, 5-phosphomevalonate kinase; MVD, mevalonate 5-diphosphate decarboxylas;DXS,1-deoxy-D-xylulose-5-phosphate synthase; DXR, 1-deoxy-D-xylulose-5-phosphate reductoisomerase; MCT, 2-C-methyl-D-erythritol-4-phosphate cytidylyltransferase; CMK, 4-diphosphocytidyl-2-C-methyl-D-erythritol kinase; MCS, 2-C-methyl-D-erythritol 2,4-cyclodiphosphate synthase; HDS,1-hydroxy-2-methyl-2-(E)-butenyl-4-diphosphate synthase; HDR, 4-hydroxy-3-methylbut-2-enyl diphosphate reductase; IDI, isopentenyl-diphosphate Delta-isomerase; GPS, geranyl diphosphate synthase; FPS, farnesyl pyrophosphate synthase.

from cholesterol into verazine was confirmed to be compromised by a series of C-22 and C-26 hydroxylation/oxidation reactions and C-26 transamination reactions in *Veratrum Californicum* and *Veratrum grandiflorum* [38, 39]. Verazine has five circular structures and has formed a basic circular skeleton. It was speculated that it is a precursor of the C-nor D-homo structure formed by the reaction of Peimine and peiminine, plant cytochrome *P450* (*CYP450*) might

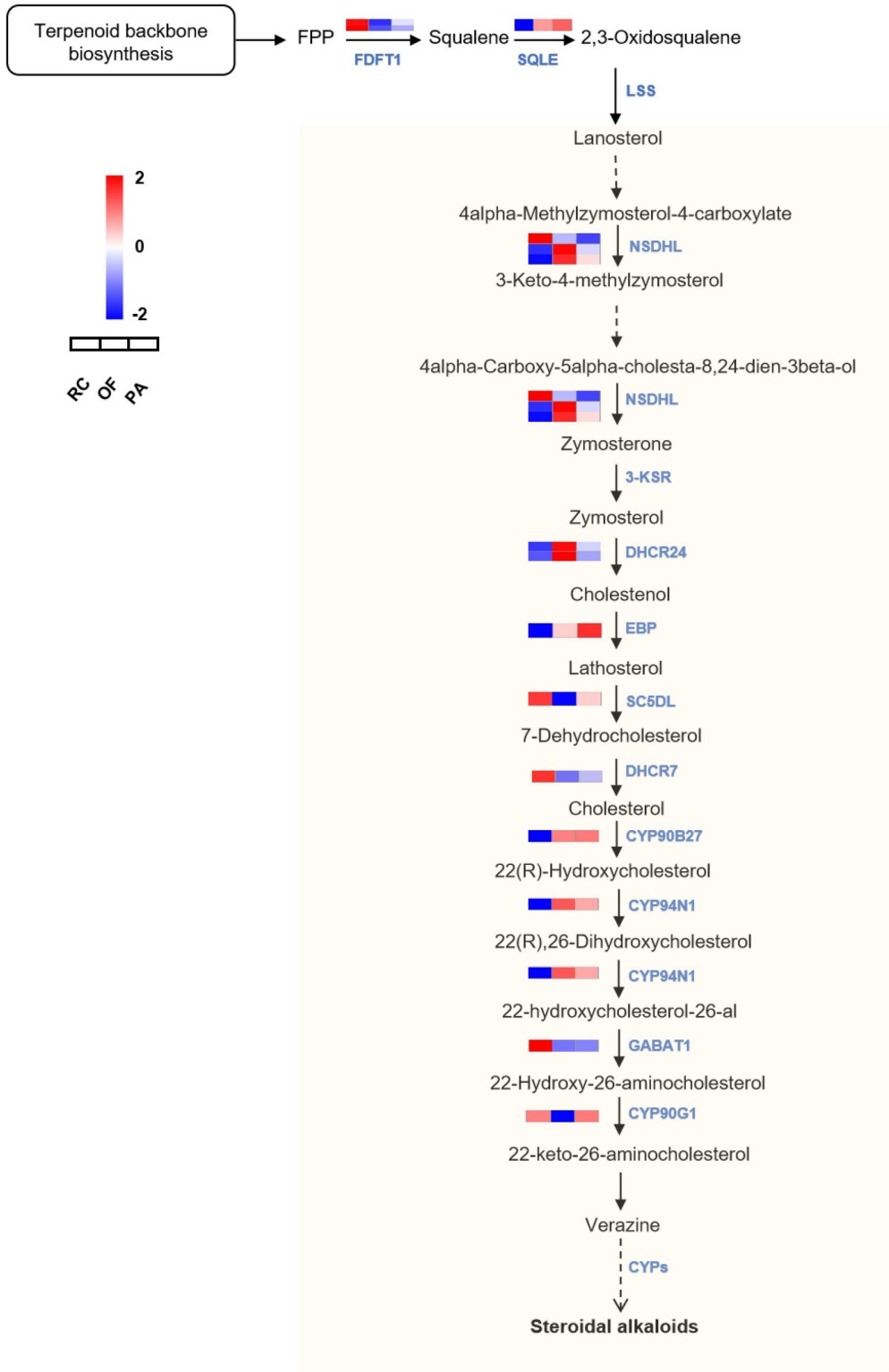

**Fig 6. Putative biosynthesis pathway from cholesterol toward steroidal alkaloids and schematic diagram of fold change in expression level of genes in the FTB of three fertilization conditions.** The dashed arrow represents the presumed terminal biosynthesis steps. FPKM of each gene were normalized according to Z-score. Red indicates high gene expression, blue indicates low gene expression. SQS, squalene synthase; SQE, squalene monooxygenase; LSS, lanosterol synthase; NSDHL, sterol-4alpha-carboxylate 3-dehydrogenase; 3-KSR, 3-keto steroid reductase; DHCR24,

delta24-sterol reductase; EBP, cholestenol Delta-isomerase; SC5DL, elta7-sterol 5-desaturase; DHCR7, 7-dehydrocholesterol reductase; CYPs, cytochrome P450s.

play prominent roles in the secondary metabolite biosynthesis processes [18]; Based on this knowledge, potential transcripts involved in steroid alkaloid biosynthesis within *F. Thunbergii* were analyzed (Fig 6, Table 3). Present speculation with different from Shan et al.'s findings [40], which speculated that peimine and peiminine in *Fritillaria anhuiensis* were biosynthesized by two precursor substances respectively. Considering the only difference in structure is C-5, we inferred that they might have common precursor cholesterol and shared some subsequent conversion reactions.

## Quantitative real-time PCR (qRT-PCR) validation

To further confirm the accuracy of the RNA-seq data, qRT-PCR analysis of 12 genes related to the steroidal alkaloid biosynthesis pathway was performed. The relative gene expression levels of *PP2C*, *SnRK2*, *ABF*, *JAZ*, *ACAT*, *DXS*, *MVD*, *FPS*, *DSDHL*, *CYP94N1*, *CYP90G1*, and *GABAT1* from three FTBs of different fertilization conditions were calculated, which were consistent with RNA-Seq FPKM values (Fig 7, S1 Table).

## Discussion

### Discrepancy of alkaloid contents in *F. Thunbergii* from different fertilization conditions

Research has shown that potassium is essential for plant growth. With sufficient nitrogen and phosphorus, more alkaloids and potassium are allocated to the underground as potassium application increases. The yield and quality of bulbs are positively correlated with the

**Table 3. Candidate CYPs associated with steroidal alkaloid biosynthesis.**

| Function | Unigenes_ID | Reference gene | Species |
|---|---|---|---|
| C-6 oxidation | Cluster-73431.46232 | *CYP85A3* [68] | *Solanum lycopersicum* |
| C-6 hydroxylation | Cluster-63258.0 | *CYP716A53v2* [68] | *Panax ginseng* |
| C-6 hydroxylation | Cluster-61668.0 | *CYP716E26* [68] | *Solanum lycopersicum* |
| C-11 hydroxylation | Cluster-73431.43439 | *CYP88D6* [68] | *Glycyrrhiza glabra* |
| C-16 hydroxylation | Cluster-73431.41324 | *CYP716A141* [64] | *Platycodon grandiflorus* |
| C-16 hydroxylation | Cluster-73431.51706 | *CYP86A2* [66] | *Arabidopsis thaliana* |
| C-20 hydroxylation | Cluster-73431.28317 | *CYP71D3* [62] | *Lotus japonicus* |
| C-22 oxidation | Cluster-73431.18721 | *CYP90G1* [38] | *Veratrum californicum* |
| C-22 hydroxylation | Cluster-73431.13425 | *CYP90B1* [54, 55] | *Arabidopsis thaliana* |
| C-22 hydroxylation | Cluster-73431.15630 | *CYP90B2* [56] | *Oryza sativa* |
| C-22 hydroxylation | Cluster-39298.0 | *CYP724B1* [56] | *Oryza sativa* |
| C-22 hydroxylation | Cluster-73431.27372 | *CYP72A188* [58] | *Solanum tuberosum* |
| C-26 oxidation | Cluster-73431.15020 | *GAME4* [62] | *Solanum lycopersicum* |
| C-26 hydroxylation | Cluster-73431.27371 | *CYP72A208* [58] | *Solanum tuberosum* |
| C-26 hydroxylation | Cluster-73431.27483 | *CYP734A6* [38, 61] | *Arabidopsis thaliana* |
| C-26 oxidation/hydroxylation | Cluster-73431.14169 | *CYP94N1* [38, 61] | *Veratrum californicum* |
| C-26 transamination | Cluster-73431.19556 | *GABAT1* [38] | *Veratrum californicum* |
| C-26 transamination | Cluster-73431.26754 | *GABAT2* [38] | *Solanum lycopersicum* |

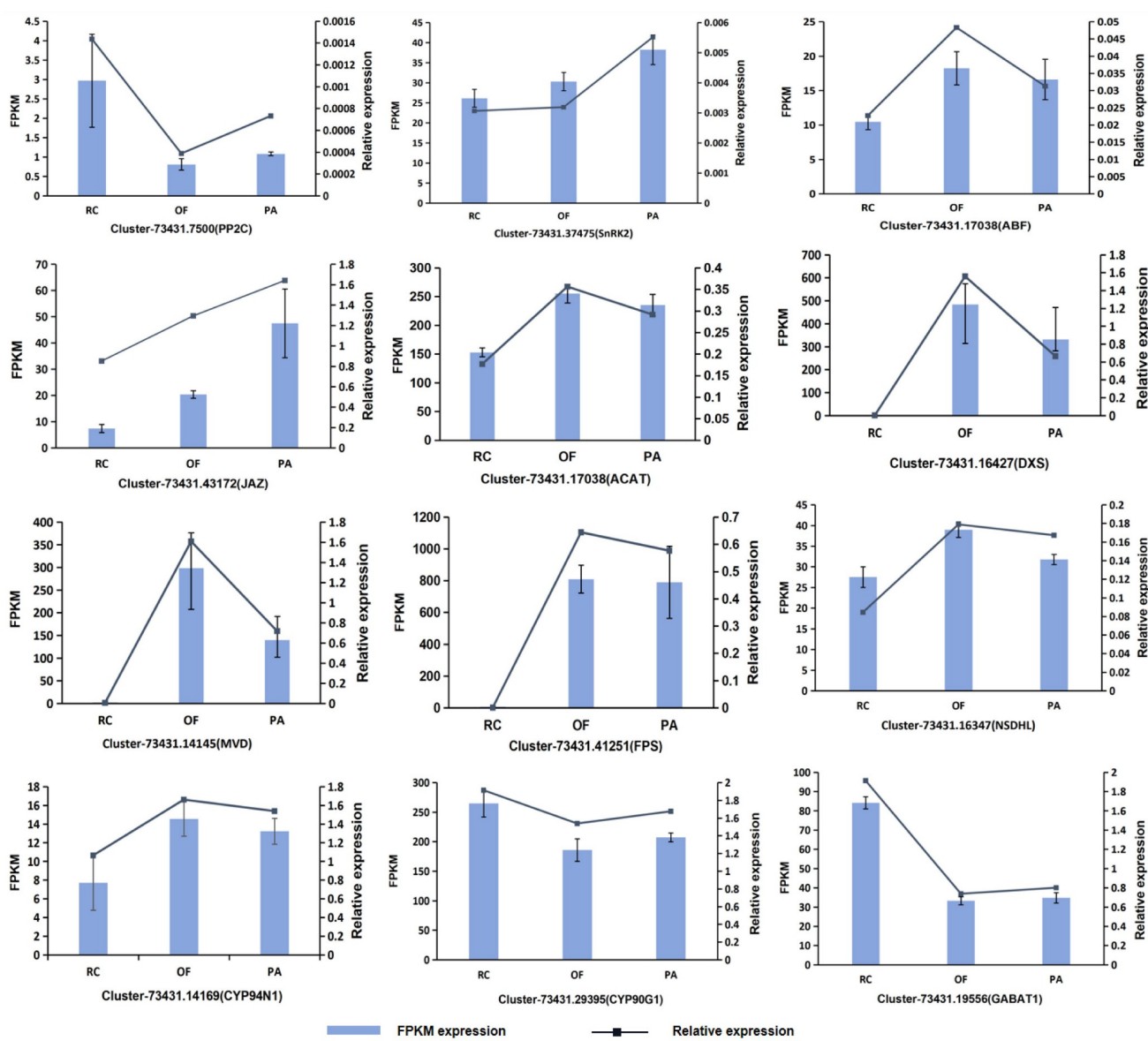

**Fig 7. Comparative expression pattern validation related to the steroidal alkaloid biosynthesis pathway structural genes as obtained from RNA-seq data and qRT-PCR.**

accumulation of underground potassium, so potassium application is an effective strategy to improve the yield and quality of *F. Thunbergii* [41, 42]. Chicken manure, organic fertilizer, and plant ash fertilizers all contain various elements necessary for plant growth. after HPLC-ELSD detection, The alkaloid content of *F. Thunbergii* under three cultivation methods in Pan'an County was detected by HPLC-ELSD. The total amount of steroidal alkaloids in FTB in RC was lower than that of other samples, but the yield under RC fertilization conditions was the highest (Table 1). The results in the plant ash group were exactly the opposite. Therefore, it is recommended to choose a cultivation method that combines organic fertilizer and plant ash, providing an experimental basis and theoretical guidance for improving yield and steroidal alkaloid content.

## Revelation of plant hormone signal transduction pathway in *F. Thunbergii*

Plant hormones are rapid response signal compounds related to regulating growth, development in key stages, and resistance to environmental stress responses [29]. According to Fig 4, 123 DEGs are distributed in eight plant hormone signal transduction pathways, among which the abscisic acid (ABA) pathway is positively correlated with the alkaloid metabolism profile of FTB, thus ABA may be the uppermost hormone for regulating the growth of FTB. ABA is the prominent medium of these response processes owing to regulating stomatal closure [43] and seed dormancy [29], resulting in the maintenance of growth and survival of plants. The gene expression of *SnRK2* and *JAZ* in PA was the highest, that *SnRK2* (ABA-dependent phosphorylase) interacts with ABA that determines the variation of symplastic calcium ion concentrations, the increase of ion efflux, the reduction of cell expansion caused by water outflow from the guard cells, and finally leading to stomatal closure [43]. *ABF* induced by ABA and collaboratively regulate *ABRE*-dependent ABA signaling involved in various stress treatments [44]. ABA was also called "stress hormone", which enhanced the stress resistance of plant tissue under environmental or abiotic stress such as cold [45] or drought [46].

Additionally, ABA had an antagonistic effect on ET, BR, and JA [47]. In the JA signaling pathway, almost all DEGs showed an upward trend in PA (Fig 4), majority of genes in the JA signaling pathway also demonstrated a similar expression spectrum. JAZ protein is a repressor of jasmonic acid (JA) signal transduction, JAZ has the highest concentration of DEGs in the JA signal transduction pathway. The gene expression level in PA samples is significantly higher than that in samples from other conditions, and the trend of gene expression in the ABA pathway is similar. Furthermore, In the BR signaling pathway, almost all DEGs showed a downward trend in PA, BR-activated transcription factors repressed the expression of desiccation transcription factor 26 (*RD26*), suggesting that antagonistic crosstalk between ABA signaling and BR signaling contributed to drought stress responses [48]. Crosstalk also existed between cytokinin and ABA, type A Arabidopsis response regulator 5 (*ARR5*) mediated by *SnRK2* is a negative regulator of cytokinin signal transduction, which heightened the stability of its protein, thus lessening cytokinin reaction during drought stress [49]. Further attempts and innovations at stress and hormone action are required to improve the principal active ingredients of *F. thunbergii*. The effect of hormones on the growth and quality of *F. thunbergii* is currently not fully studied, and further experimental verification will be designed in the future.

## Speculation of plant steroid alkaloids biosynthesis associated genes

Based on the present data, we screened out candidate genes that affected the accumulation of steroidal alkaloids in FTB, which could help improve the quality of *F. thunbergii*. Firstly, in the biosynthesis of terpenoid backbone, as though both MVA and MEP pathways existed in plants, only the genes in the MEP pathway of *F. thunbergii* are higher than *F. cirrhosae*, which indicating the MEP pathway may be the main way to control the biosynthesis of DMAPP/IPP [50]. We found that the DEGs were mainly distributed in MEP pathways, which was consistent with previous conclusions (Fig 5, S8 Table). *DXS* is the first rate-limiting step in the catalytic MEP pathway, which catalyzes the production of 1-deoxyglucose 5-phosphate *(DXP)* from pyruvate and 3-phosphate glyceraldehyde [51]. *DXR* is the second-rate limiting enzyme in the MEP pathway, which can catalyze the synthesis of various important intermediate metabolite enzymes such as IPP, pyridoxine, and thiamine by *DXP* [52]. Compared with RC samples, the expression level of *DXS* and *DXR* in PA and OF exceeded RC, which provided strong evidence for the enzyme to be the key rate-limiting enzyme for the biosynthesis of the terpenoid backbone.

Understanding the medicinal ingredients biosynthesis of FTB is essential to bioengineering and metabolic engineering for sustainable production [38, 53]. So far, we do not know the specific mechanism of the downstream pathway from cholesterol toward steroidal alkaloids in FTB, and the pathway prediction diagram was plotted based on previous literature (Fig 6). A large number of enzymes from other plants took part in the switch from cholesterol to steroidal alkaloids or glycoalkaloids have been reported. The steroid hydroxylation at C-22 position could be catalysed by *CYP90B27* in *Veratrum californicum* [38], *CYP90B1* (*DWF4*) in *Arabidopsis thaliana* [54, 55], *CYP90B2* and *CYP724B1* in *Oryza sativa* (rice) [56], *CYP90B3* and *CYP724B2* in *Lycopersicon esculentum* (tomato) [57], and *CYP72A188* (*PGA2*) in *Solanum tuberosum* (potato) [58]. Next, *CYP734A1* in *A. thaliana* [59], *CYP734A7* in *L. esculentum* [60], and *CYP72A208* (*PGA1*) in *S. tuberosum* [58] participated in C-26 hydroxylation reactions. In addition, *CYP94N1* in *V. californicum* and *CYP734A6* in *O. sativa* also participated in C-26 oxidation [38, 61]. Afterward, the products of the previous reactions were subject to transamination reaction at C-26 by *GABAT1* [38] or *GAME12* [62] and oxidation at C-22 by *CYP94N1* or *CYP90G1* [38], which is prepared for the formation of F-ring and the generation of verazine. The previous research revealed that cevanine, veratramine, and jervin were probably generated by the same C-nor-D-homo-steroidal intermediate [63], but some specific enzymes catalyzed the subsequent reaction of conversion from verazine to C-nor-D-homo-steroidal skeleton is still unknown. The synthesis mechanism may mainly include C-16 hydroxylation, C-22 reduction, wagner-meerwein rearrangement, and C-12 hydroxylation reactions. The pyrrolidine ring-opening reaction and cyclization reaction might be carried out again later [39] (Fig 6). Additionally, the ability to catalyze C-16 hydroxylation was also enlightened for *CYP716s*, such as *PgCYP716A141* [64], *BfCYP716AY1* [65], *AtCYP86A2* [66] and *St16DOX* [67]. *CYP716A47* in *Panax ginseng* partake in C-12 hydroxylation [68]. According to the structure of peimine and peiminine, it is possible to proceed with hydroxylation reactions at C-6 and C-20. The expression level of genes is related to the accumulation of alkaloids in *F. thunbergii*, but the functional function of related genes still needs further verification.

## Conclusion

Different fertilization conditions have different effects on the yield and quality of *F. thunbergii*. This study revealed the potential mechanism of increased steroidal alkaloid content in the bulb yield of Fritillaria after applying different fertilizers. According to the analysis of differences in steroidal alkaloid content and gene expression levels, The PA and OF samples had multiple common genes upregulated in the plant hormone signal transduction pathway, terpenoid skeleton biosynthesis pathway, and cholesterol synthesis pathway, while RC samples showed the opposite trend. Within plant hormone signal transduction pathways, the ABA signal transduction pathway is positively correlated with the steroid alkaloid metabolism spectrum of *F. thunbergii*, suggesting that ABA might be the most important hormone regulating the synthesis of steroid alkaloids in *F. thunbergii*. Furthermore, compared with the MVA pathway, the MEP pathway contained a higher number of DEGs from different conditions, which might be a key synthesis pathway in *F. thunbergii* that affects the accumulation of downstream steroidal alkaloids by controlling DMAPP/IPP biosynthesis. Cholesterol has been characterized as a precursor substance of steroid alkaloids, however, the downstream biosynthesis of cholesterol to steroidal alkaloids in plants remained unclear. Conduct in-depth exploration of key genes in the biosynthesis pathway of steroidal alkaloids, we identified a total of 62 differentially expressed genes, including 9 of MVA pathways and 29 of MEP pathways, and the expression trend of 19 differentially expressed genes in the

steroid synthesis pathway are consistent with the accumulation of steroid alkaloids, which may be involved in steroid alkaloid synthesis, laying the foundation for elucidating the biosynthetic pathway of steroid alkaloids in Streptomyces.

## Supporting information

**S1 Fig. PCA principal component analysis diagram.**
(TIF)

**S2 Fig. Functional annotations and classification of the assembled unigenes.** (A) Homologous species distribution unigenes about FTB compared against the Nr database. (B) GO function annotation and classification statistics of the assembled unigenes. The results are summarized in three main categories: BP, biological process; CC, cellular component; MF, molecular function. (C) KEGG function annotation and classification statistics of the assembled unigenes.
(ZIP)

**S1 Table. Primers for qRT-PCR analysis.**
(DOCX)

**S2 Table. Numbers and quality of RNA-seq reads produced in each sample.**
(DOCX)

**S3 Table. Length distribution of unigenes from the transcriptome data.**
(DOCX)

**S4 Table. Transcripts and FPKM of genes involved in plant hormone signal transduction (ko04075).**
(DOCX)

**S5 Table. Transcripts and FPKM of genes involved in plant-pathogen interaction (ko04626).**
(DOCX)

**S6 Table. Transcripts and FPKM of genes involved in mRNA surveillance pathway (ko03015).**
(DOCX)

**S7 Table. Transcripts and FPKM of genes involved in ubiquitin mediated proteolysis (ko04120).**
(DOCX)

**S8 Table. Transcripts and FPKM of genes involved in terpenoid backbone biosynthesis (ko00900).**
(DOCX)

**S9 Table. Transcripts and FPKM of genes involved in steroid biosynthesis (ko00100).**
(DOCX)

## Author Contributions

**Data curation:** Shuang Liang, Hongwei Fu.

**Formal analysis:** Lei Luo.

**Software:** Mengmin Wu.

**Supervision:** Hongwei Fu.

**Validation:** Hongwei Fu.

**Writing – original draft:** Luman Huang.

**Writing – review & editing:** Zhuoheng Zhong.

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
