## [Decision Letter · Decision Letter 0]

10 Jun 2024

PONE-D-24-13936Transcriptomic analysis reveals effects of fertilization towards growth and quality of Fritillariae thunbergii  bulbusPLOS ONE

Dear Dr. Zhong,

Thank you for submitting your manuscript to PLOS ONE. After careful consideration, we feel that it has merit but does not fully meet PLOS ONE’s publication criteria as it currently stands. Therefore, we invite you to submit a revised version of the manuscript that addresses the points raised during the review process.

Thank you for your contributions to the field. Suggest making revisions based on the reviewer's comments. All my decisions are made in accordance with PLOS ONE publication standards.

We look forward to receiving your revised manuscript.

Kind regards,

Minhui Li, PhD

Academic Editor

PLOS ONE

3. PLOS requires an ORCID iD for the corresponding author in Editorial Manager on papers submitted after December 6th, 2016. Please ensure that you have an ORCID iD and that it is validated in Editorial Manager. To do this, go to ‘Update my Information’ (in the upper left-hand corner of the main menu), and click on the Fetch/Validate link next to the ORCID field. This will take you to the ORCID site and allow you to create a new iD or authenticate a pre-existing iD in Editorial Manager. Please see the following video for instructions on linking an ORCID iD to your Editorial Manager account: https://www.youtube.com/watch?v=_xcclfuvtxQ.

Additional Editor Comments:

Two reviewers have provided valuable feedback. Please respond to their comments and make appropriate additions to the article.

Additional Comments from PLOS Editorial Office:

We note that one or more reviewers has recommended that you cite specific previously published works. As always, we recommend that you please review and evaluate the requested works to determine whether they are relevant and should be cited. It is not a requirement to cite these works. We appreciate your attention to this request.

Reviewers' comments:

Reviewer's Responses to Questions

**Comments to the Author**

1. Is the manuscript technically sound, and do the data support the conclusions?

Reviewer #1: Partly

Reviewer #2: Yes

2. Has the statistical analysis been performed appropriately and rigorously? 

Reviewer #1: I Don't Know

Reviewer #2: Yes

3. Have the authors made all data underlying the findings in their manuscript fully available?

Reviewer #1: Yes

Reviewer #2: Yes

4. Is the manuscript presented in an intelligible fashion and written in standard English?

Reviewer #1: Yes

Reviewer #2: Yes

5. Review Comments to the Author

Reviewer #1: The Abstract is lacking key results like the yield and alkaloids content obtained from the best fertilizer.

In the introduction, is is preferable to state the hypothesis tested rather than the objectives.

In the methods section, the authors should justify why they use only one cultiver rather than using many. the quantity and justification of the quantity of eah of the fertiliser used for the study. what was the growing conditions of the plants. Alkaloids content was evaluated of plant tissue of how old? You should use AOVA test with mean separation as a statistical means. Also provide The software used for clustering transcriptome data

Reviewer #2: The manuscript entitled "Transcriptomic analysis reveals effects of fertilization towards growth and quality of Fritillariae thunbergii bulbus" provides valuable insights into optimizing the cultivation conditions of Fritillariae thunbergii (FTB), a traditional Chinese medicine renowned for its medicinal properties. The study investigates the impact of different fertilizers on the yield and quality of FTB, focusing on raw chicken manure (RC), organic fertilizer (OF), and plant ash (PA). Through a combination of HPLC-ELSD detection and yield statistics, OF application emerged as the most effective fertilizer. Transcriptome analysis uncovered the up-regulation of the ABA signaling pathway, potentially enhancing bulb yield, along with the identification of key genes associated with steroidal alkaloid accumulation. However, the manuscript requires minor revisions. Specifically, replication information regarding biological and technical replicates should be provided, and transcription factor/gene names should be italicized throughout the manuscript. Additionally, the methodology section should include NCBI SRA submission information. Furthermore, the introduction and discussion sections would benefit from the inclusion of references such as (Kumar, V., Sharma, S., Kumar, P. (2024). Insight into the Genetics and Genomics Studies of the Fritillaria Species. In: Gahlaut, V., Jaiswal, V. (eds) Genetics and Genomics of High-Altitude Crops. Springer, Singapore. https://doi.org/10.1007/978-981-99-9175-4_4;
https://doi.org/10.1016/j.indcrop.2023.117541; Kumar, V., Kumar, P., Bhargava, B. et al. Transcriptomic and Metabolomic Reprogramming to Explore the High-Altitude Adaptation of Medicinal Plants: A Review. J Plant Growth Regul 42, 7315–7329 (2023). https://doi.org/10.1007/s00344-023-11018-8; Kapoor, B., Kumar, A. & Kumar, P. Transcriptome repository of North-Western Himalayan endangered medicinal herbs: a paramount approach illuminating molecular perspective of phytoactive molecules and secondary metabolism. Mol Genet Genomics 296, 1177–1202 (2021). https://doi.org/10.1007/s00438-021-01821-x) to enrich the context and support the findings. Overall, these adjustments will enhance the clarity and completeness of the manuscript, contributing to its scientific significance.

6. PLOS authors have the option to publish the peer review history of their article (what does this mean?). If published, this will include your full peer review and any attached files.

Reviewer #1: **Yes: **Eric Bertrand Kouam

Reviewer #2: No

---

## [Author Response · Author response to Decision Letter 0]

11 Jul 2024

Dear Editor and Reviewers:

Thanks for your letter and the reviewers' comments concerning our manuscript, "Transcriptomic analysis reveals effects of fertilization towards growth and quality of Fritillariae thunbergii bulbus" (Manuscript ID: PONE-D-24-13936). Those comments are all valuable and helpful for revising and improving our paper. We have studied all comments carefully and have made conscientious corrections. Our response is given in standard font, and changes to the manuscript are given in red text. The responses to the reviewers' comments are as follows:

Response to the reviewers:

Reviewer 1

We feel great thanks for your professional review work on our article. As you are concerned, there are several problems that need to be addressed. According to your nice suggestions, we have made extensive corrections to our previous draft, the detailed corrections are listed below.

1. The Abstract is lacking key results like the yield and alkaloids content obtained from the best fertilizer.

Answer: Line 25-27, the yield and alkaloids content of “The combined results of HPLC-ELSD detection and yield statistics showed that the F. thunbergii with OF application was the best, with the content of peimine and peiminine reaching 0.0603% and 0.0502%, respectively. In addition, the yield was 2.70 kg/m². ” was added.

2. In the introduction, it is preferable to state the hypothesis tested rather than the objectives.

Answer: Line 71-77, and Line 82-84, based on existing literatures on the correlation between altitude and secondary metabolic yield of medicinal plants, and the application to in vitro cultivation of Himalayan Fritillaria species, it is speculated that different fertilization methods may be related to the alkaloid content and yield of Fritillaria.

3. In the methods section, the authors should justify why they use only one cultiver rather than using many. the quantity and justification of the quantity of each of the fertiliser used for the study. what was the growing conditions of the plants. Alkaloids content was evaluated of plant tissue of how old? You should use AOVA test with mean separation as a statistical means. Also provide The software used for clustering transcriptome data.

Answer: To control variables, we selected the same batch of Fritillariae thunbergii bulbs for different fertilization conditions to prevent the influence of different varieties on their alkaloid contents. Among them, ‘Zhebei 1’ has a wide range of applications, which is more representative and practical.

The growth environment of Fritillariae thunbergii bulbs in the three experimental fields was the same natural environment, with the same altitude 600 m.

Three year old bulbs of Fritillariae thunbergii were used in this experiment. 

Line 173-174, and Line 194-196, we conducted One-way ANOVA of alkaloid content of different fertilization conditions using SPSS (25.0.0). 

Line 161-162, “Furthermore, the clustered expression pattern was drawn through R (Version 3.0.3), Package ggplot2 and Package pheatmap.” were added.

Additionally, Line 184-193, we have provided a more detailed description of the experimental results.

4.There is no standard deviation?- yield

Answer: Thank you for your reminder. Line 93-94, Fritillaria thunbergii bulb yield was determined by harvesting the plants in an area of 2 m2 in each plot. We collected bulbs from a whole experimental field and calculated the yield.

Reviewer 2

The manuscript entitled "Transcriptomic analysis reveals effects of fertilization towards growth and quality of Fritillariae thunbergii bulbus" provides valuable insights into optimizing the cultivation conditions of Fritillariae thunbergii (FTB), a traditional Chinese medicine renowned for its medicinal properties. The study investigates the impact of different fertilizers on the yield and quality of FTB, focusing on raw chicken manure (RC), organic fertilizer (OF), and plant ash (PA). Through a combination of HPLC-ELSD detection and yield statistics, OF application emerged as the most effective fertilizer. Transcriptome analysis uncovered the up-regulation of the ABA signaling pathway, potentially enhancing bulb yield, along with the identification of key genes associated with steroidal alkaloid accumulation. However, the manuscript requires minor revisions.

Answer: Thank you for your valuable comments, revisions have been made to improve the readability of the present manuscript. Detailed changes and corresponding answers have been displayed below.

1. Replication information regarding biological and technical replicates should be provided.

Answer: Line 95-96, “ all the bulbs were collected biological replicates to randomly select different regions in the north, south, east, and west of the same production area” was added. Line 99-100, and Line 112-113, the number biological replicates has been explained.

2. Transcription factor/gene names should be italicized throughout the manuscript.

Answer: We were really sorry for our careless mistakes. Thank you for noticing. Transcription factor/gene names were corrected to Italic format.

3. The methodology section should include NCBI SRA submission information.

Answer: Line 175-177, “Data avaliability: The raw data of RNA-Seq have been submitted to the the NCBI Sequence Read Archive (SRA) database (Accession: PRJNA950562).” was added.

4. The introduction and discussion sections would benefit from the inclusion of references such as (Kumar, V., Sharma, S., Kumar, P. (2024). Insight into the Genetics and Genomics Studies of the Fritillaria Species. In: Gahlaut, V., Jaiswal, V. (eds) Genetics and Genomics of High-Altitude Crops. Springer, Singapore. https://doi.org/10.1007/978-981-99-9175-4_4;https://doi.org/10.1016/j.indcrop.2023.117541; Kumar, V., Kumar, P., Bhargava, B. et al. Transcriptomic and Metabolomic Reprogramming to Explore the High-Altitude Adaptation of Medicinal Plants: A Review. J Plant Growth Regul 42, 7315–7329 (2023). https://doi.org/10.1007/s00344-023-11018-8; Kapoor, B., Kumar, A. & Kumar, P. Transcriptome repository of North-Western Himalayan endangered medicinal herbs: a paramount approach illuminating molecular perspective of phytoactive molecules and secondary metabolism. Mol Genet Genomics 296, 1177–1202 (2021). https://doi.org/10.1007/s00438-021-01821-x) to enrich the context and support the findings. Overall, these adjustments will enhance the clarity and completeness of the manuscript, contributing to its scientific significance.

Answer: We sincerely appreciate the valuable comments. We have checked the literature carefully and added the four references on 11 (Line 58-60), 15 (Line 71-72), 16 (Line 71-72) and 17 (Line 72-73) into the Introduction part in the revised manuscript.

Thank you very much for your attention and time. Look forward to hearing from you.

Yours sincerely,

Corresponding author: 

Name: Zhuoheng Zhong

E-mail: zhongzhh@zstu.edu.cn

---

## [Decision Letter · Decision Letter 1]

22 Aug 2024

Transcriptomic analysis reveals effects of fertilization towards growth and quality of Fritillariae thunbergii  bulbus

PONE-D-24-13936R1

Dear Dr. Fu,

We’re pleased to inform you that your manuscript has been judged scientifically suitable for publication and will be formally accepted for publication once it meets all outstanding technical requirements.

Kind regards,

Minhui Li, PhD

Academic Editor

PLOS ONE

Additional Editor Comments (optional):

Thank you for your contributions to the field.

Reviewers' comments:

Reviewer's Responses to Questions

**Comments to the Author**

1. If the authors have adequately addressed your comments raised in a previous round of review and you feel that this manuscript is now acceptable for publication, you may indicate that here to bypass the “Comments to the Author” section, enter your conflict of interest statement in the “Confidential to Editor” section, and submit your "Accept" recommendation.

Reviewer #1: All comments have been addressed

Reviewer #2: All comments have been addressed

Reviewer #3: All comments have been addressed

2. Is the manuscript technically sound, and do the data support the conclusions?

Reviewer #1: Yes

Reviewer #2: Yes

Reviewer #3: Yes

3. Has the statistical analysis been performed appropriately and rigorously? 

Reviewer #1: Yes

Reviewer #2: Yes

Reviewer #3: Yes

4. Have the authors made all data underlying the findings in their manuscript fully available?

Reviewer #1: Yes

Reviewer #2: Yes

Reviewer #3: Yes

5. Is the manuscript presented in an intelligible fashion and written in standard English?

Reviewer #1: Yes

Reviewer #2: Yes

Reviewer #3: Yes

6. Review Comments to the Author

Reviewer #1: Susbtancial improvement has been carried out in the revised manuscript. I am satisfied with the answers provided to the different questions that was raised.

Reviewer #2: The authors have carefully addressed the comments and duly incorporated in the revised manuscript. Hence manuscript can be accepted in its current form.

Reviewer #3: The author has revised the paper as requested and responded to the questions. I agree to publish this manuscript in Plos one

7. PLOS authors have the option to publish the peer review history of their article (what does this mean?). If published, this will include your full peer review and any attached files.

Reviewer #1: **Yes: **Eric Bertrand Kouam

Reviewer #2: No

Reviewer #3: **Yes: **Jingmao You

---

## [Editor Report · Acceptance letter]

10 Sep 2024

PONE-D-24-13936R1 

PLOS ONE

Dear Dr. Fu, 

I'm pleased to inform you that your manuscript has been deemed suitable for publication in PLOS ONE. Congratulations! Your manuscript is now being handed over to our production team.

Kind regards, 

on behalf of

Dr. Minhui Li 

Academic Editor

PLOS ONE